# Epistasis lowers the genetic barrier to SARS-CoV-2 neutralizing antibody escape

Leander Witte [1,4], Viren A. Baharani[1,2,4], Fabian Schmidt[1,4], Zijun Wang[2], Alice Cho[2], Raphael Raspe [2], Camila Guzman-Cardozo[1], Frauke Muecksch[1], Marie Canis[1], Debby J. Park[1], Christian Gaebler [2], Marina Caskey [2], Michel C. Nussenzweig [2,3] ✉, Theodora Hatziioannou [1] ✉ & Paul D. Bieniasz [1,3] ✉

Waves of SARS-CoV-2 infection have resulted from the emergence of viral variants with neutralizing antibody resistance mutations. Simultaneously, repeated antigen exposure has generated affinity matured B cells, producing broadly neutralizing receptor binding domain (RBD)-specific antibodies with activity against emergent variants. To determine how SARS-CoV-2 might escape these antibodies, we subjected chimeric viruses encoding spike proteins from ancestral, BA.1 or BA.2 variants to selection by 40 broadly neutralizing antibodies. We identify numerous examples of epistasis, whereby in vitro selected and naturally occurring substitutions in RBD epitopes that do not confer antibody resistance in the Wuhan-Hu-1 spike, do so in BA.1 or BA.2 spikes. As few as 2 or 3 of these substitutions in the BA.5 spike, confer resistance to nearly all of the 40 broadly neutralizing antibodies, and substantial resistance to plasma from most individuals. Thus, epistasis facilitates the acquisition of resistance to antibodies that remained effective against early omicron variants.

Over the course of the COVID-19 pandemic, successive waves of SARS-CoV-2 infection have been driven in part by the repeated emergence of variants with mutations that confer resistance to neutralizing antibodies[1–5]. Of particular note, the omicron BA.1, BA.2 variants and derivatives thereof, exhibited a substantial and previously unprecedented loss of sensitivity to plasma antibodies elicited by vaccination, or infection with prior variants[3–5].

Antibodies targeting the SARS-CoV-2 RBD are the dominant source of neutralizing activity in the plasma of infected or vaccinated individuals[6,7] and most RBD antibodies can be broadly grouped into 4 prototype classes[8,9]. Class 1 and 2 antibodies bind epitopes overlapping the ACE2 binding site, while class 3 and 4 antibodies bind outside the ACE2-binding site on opposite sides of the RBD. The class 4 antibody epitope exhibits greater sequence conservation than other epitopes, but is concealed when the RDB is in the 'down' conformation.

The numerous RBD substitutions that were acquired by the omicron BA.1 and BA.2 variants conferred resistance to many individual antibodies that have been cloned from vaccinated or infected individuals. Omicron emergence correspondingly reduced the effectiveness of RBD-binding neutralizing antibodies as components of the convalescent and vaccine-elicited immune responses, and as therapeutics[3–5,10–12].

Nevertheless, through prolonged and/or repeated exposure to the spike antigen, human SARS-CoV-2-neutralizing antibodies have evolved and diversified through affinity maturation[13–18]. This process has enabled a subset of more broadly neutralizing antibodies to acquire resilience to spike substitutions that would otherwise confer resistance, and thereby retain neutralizing activity against divergent variants such as BA.1 and BA.2[3,16,19,20]. The presence of such broadly neutralizing antibodies in plasma and in memory B cells likely

[1]Laboratory of Retrovirology, The Rockefeller University, New York, NY 10065, USA. [2]Laboratory of Molecular Immunology, The Rockefeller University, New York, NY 10065, USA. [3]Howard Hughes Medical Institute, The Rockefeller University, New York, NY 10065, USA. [4]These authors contributed equally: Leander Witte, Viren A. Baharani, Fabian Schmidt. ✉e-mail: nussen@rockefeller.edu; thatzio@rockefeller.edu; pbieniasz@rockefeller.edu

contributes to the residual effectiveness of ancestral variant-based vaccines against infection and particularly against serious disease. A key question, therefore, is whether and how SARS-CoV-2 might evolve to evade these evolved, broadly neutralizing antibodies.

In this work, we applied selection pressure, using a collection of 40 such antibodies to ancestral Wuhan-Hu-1, as well as BA.1 or BA.2 spike sequences, to generate resistant variants. These analyses reveal frequent epistasis between pre-existing substitutions in BA.1/BA.2 and resistance mutations acquired during selection experiments, some of which correspond to substitutions appearing in BA.2 daughter lineages. Resistance to broadly neutralizing antibodies is more readily acquired by BA.1/BA.2 than ancestral lineages - single substitutions that confer resistance to more than half of the antibodies in the context of BA.1 or BA.2 fail to do so in the context of Wuhan-Hu-1. Moreover, only two or three of these substitutions introduced into the context of BA.5 are sufficient to confer resistance to nearly all broadly neutralizing antibodies tested. Our findings suggest that epistatic interaction between pre-existing and emergent substitutions in neutralizing epitopes could substantially affect the direction of future SARS-CoV-2 evolution, with implications for vaccine immunogen effectiveness and design.

## Results

### Selection of SARS-CoV-2 spike variants with antibodies

We chose 40 antibodies, isolated from the memory B cells of multiple volunteer cohorts with varying exposures to SARS-CoV-2 antigens, using an ancestral Wuhan-Hu-1 RBD (Supplementary Data 1 and 2). Specifically, the antibodies were from (i) participants infected with Wuhan-Hu-1 at 6.2 or 12 months previously, some of whom had also received 1 or 2 mRNA vaccine doses prior to antibody isolation[13,17] (ii) recipients of 3 mRNA vaccine doses who had not been infected (antibodies isolated at ~1 m after the third dose)[16] or (iii) individuals who had experienced an omicron BA.1 breakthrough infection after 3 vaccine doses (antibodies isolated at ~1 m after infection)[21,22]. We chose 'broadly' neutralizing antibodies, defined as those that could neutralize both Wuhan-Hu-1 and omicron (BA.1) pseudotyped viruses, with

IC$_{50}$ values of ~100 ng/ml or less. Most of these antibodies (35 of 40) also neutralized BA.2. With the exception of two class 1/4 broadly neutralizing antibodies, isolated from infected individuals at 1.3 m post-infection[9,23], the antibodies were chosen without regard to any property (e.g., V$_H$, V$_L$ or RBD epitope class, Supplementary Data 2) other than potency and breadth. Based on competition experiments with prototype class 1 through 4 antibodies, the broadly neutralizing collection targeted diverse RBD epitopes that included all classes, but exhibited some skew toward class 3.

To select antibody escape mutants, we employed replication-competent recombinant vesicular stomatitis viruses (rVSV/SARS-CoV-2)[24] that encoded the spike proteins from ancestral Wuhan-Hu-1, BA.1 or BA.2 variants in place of the VSV G protein. Diversified rVSV/SARS-CoV-2 populations containing 10$^6$ infectious units were incubated with antibodies at 1 μg/ml before infection of target cells. Following 2 passages in the presence of each antibody, the acquisition of RBD substitutions in the selected progeny viral populations was evaluated by next-generation sequencing (Supplementary Data 3). There was no apparent difference in the ability of antibodies from different classes to select mutations. Indeed, in a total of 120 selection experiments, 39 out of the 40 antibodies tested yielded substitutions, present at a frequency of >10% of sequences in the selected viral populations. Overall, these substitutions were at 34 different positions in the RBD (Fig. 1a, b, Supplementary Fig. 1). Substituted positions in the selected viral populations were spatially proximal to, or within, the ACE2 binding site and were largely consistent with the selecting antibody class designation, determined by competition with prototype class 1–4 antibodies (Supplementary Data 2, Supplementary Fig. 1). With only 3 exceptions, two of which were reversion of BA.1 specific substitution (F375S and S496G), the antibody selection experiments in the BA.1 and BA.2 context generated substitutions at positions that were not already changed in BA.1 or BA.2 compared to Wuhan-Hu-1 (Fig. 1a, b, Supplementary Fig. 1).

### Broadly neutralizing antibody resistance mutations

We chose 23 substitutions that were enriched during the passage of rVSV/SARS-CoV-2 variants in the presence of the 40 antibodies and

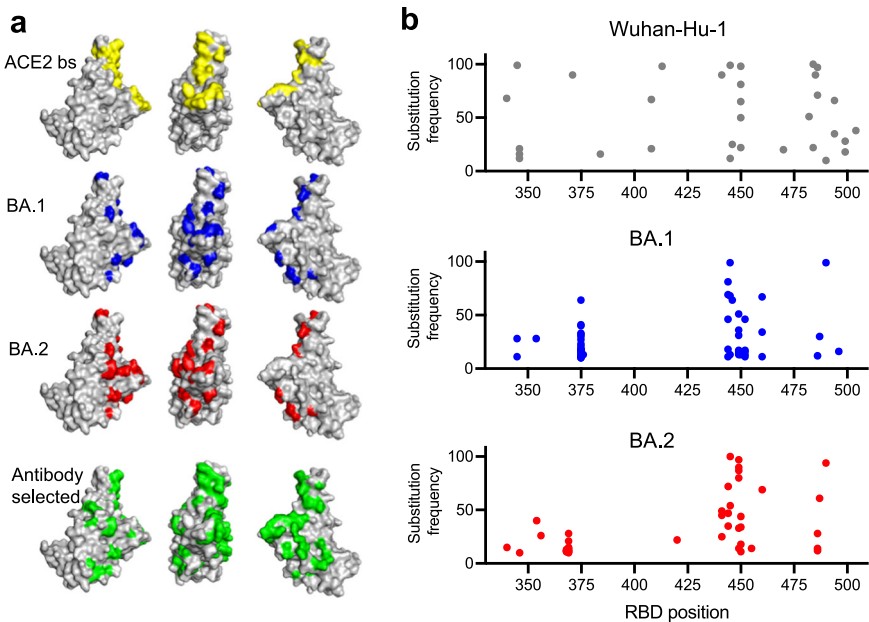

**Fig. 1 | Selection of SARS-CoV-2 spike variants with antibodies. a** Receptor-Binding Domain (RBD) structure (PDB ID: 7C8J [https://www.rcsb.org/structure/7c8j]) illustrating salient features: yellow = ACE2-binding site; blue = residues changed in BA.1 relative to Wuhan-Hu-1; red = residues changed in BA.2. Highlighted in green are all positions where substitutions were selected by neutralizing antibodies during replication of rVSV/SARS-CoV-2 encoding Wuhan-Hu-1, BA.1, or BA.2 spike proteins. **b** Substitution frequency at residues along the length of the RBD following passage of rVSV/SARS-CoV-2 encoding Wuhan-Hu-1, BA.1, and BA.2 spike proteins with antibodies.

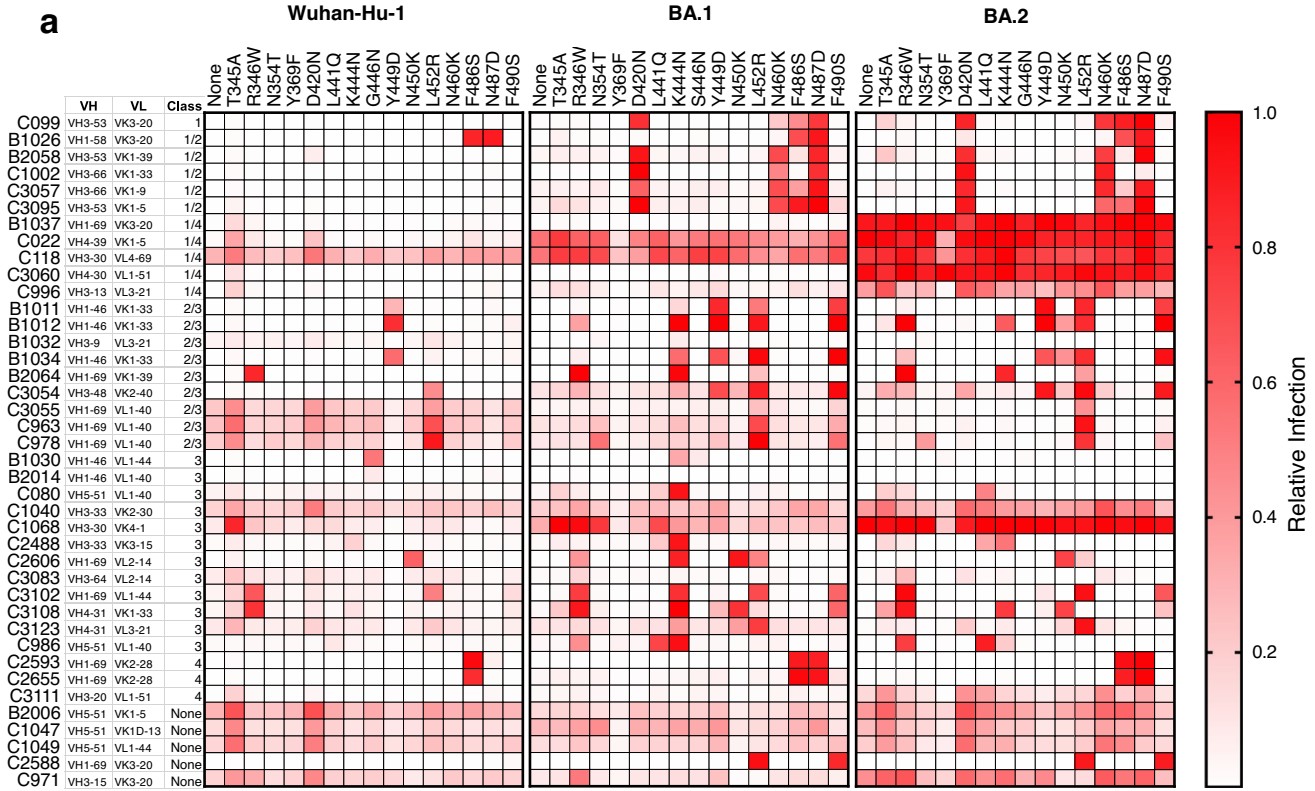

**Fig. 2 | Effects of substitutions in Wuhan-Hu-1, BA.1, and BA.2 backgrounds on neutralizing antibody sensitivity. a** Inhibition of Wuhan-Hu-1, BA.1, and BA.2 receptor-binding domain (RBD) point mutant pseudotypes by broadly neutralizing antibodies. Relative infection is defined as the decimal fraction of infection measured (with 1 μg/ml antibody), relative to an uninhibited virus control (without antibody). Median value of two experiments.

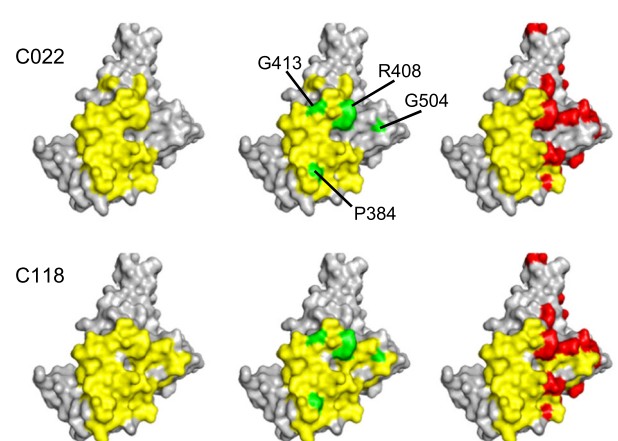

**Fig. 3 | Resistance to broadly neutralizing class 4 and 1/4 antibodies.** Receptor-binding domain (RBD) structure (PDB ID: 7C8J [https://www.rcsb.org/structure/7c8j]) illustrating epitopes of two prototype class 4 antibodies (C022 and C118, yellow), substitutions that confer resistance to class 4 and 1/4 antibodies (green), and pre-existing substitutions in BA.2 (red).

constructed spike expression plasmids based on the context in which the substitutions were selected. If a substitution was enriched by antibody selection in the ancestral rVSV/SARS-CoV-2$_{Wuhan-Hu-1}$ context, a corresponding mutant Wuhan-Hu-1 spike expression plasmid was constructed. In instances where a position was substituted in either of the antibody selected rVSV/SARS-CoV-2$_{BA.1}$ or rVSV/SARS-CoV-2$_{BA.2}$ populations, the corresponding position was substituted in all three parental spike expression plasmids. Thus, we generated 56 pseudotyped HIV-1 viruses with mutant spike sequences and tested their

ability to resist neutralization by each of the 40 broadly neutralizing antibodies, at a concentration of 1 μg/ml (Fig. 2, Supplementary Data 4). Infection was quantified relative to uninhibited virus (absence of antibody), and antibody 'escape' was defined as a fivefold increase in infection relative to the antibody-inhibited parental pseudotype, provided infection reached >10% of the uninhibited control. Most of the substitutions conferred specific resistance to one or more antibodies of the same class, consistent with their position on the RBD surface. For example, class 4 or 1/4 broadly neutralizing antibodies, which bind epitopes that are well conserved in sarbecoviruses[9] were escaped by mutations at P384, R408, G413, or G504 (Fig. 3, Supplementary Data 4, Supplementary Fig. 1). Notably, some class 1/4 or 4 broadly neutralizing antibodies were ineffective or less potent against BA.2, which has substitutions that encroach on class 4 epitopes (Fig. 2 and 3). While most substitutions resulted in class-specific antibody resistance, two substitutions positioned at the base of the RBD either gave modest class-independent reductions in sensitivity (F375S), or no reductions in antibody sensitivity (Y369F). We speculate that these represent fitness-enhancing substitutions that arose during the rVSV/SARS-CoV-2 passage. In the case of Y369F, which arose in selection experiments with diverse class 1, 1/2, 2/3, or 3 antibodies, there was an increase in sensitivity to some antibodies (Fig. 2), especially those in class 4, that is ordinarily concealed when the RBD is in the 'down' configuration. Y369 falls within the epitope of class 4 antibodies, and it is possible that this substitution sensitizes the spike to these antibodies by directly improving antibody binding affinity, or by increasing exposure to the class 4 antibody epitope.

## Epistasis and neutralizing antibody resistance

For more than half of the broadly neutralizing antibodies (21 out of 40), we found single-amino-acid substitutions that conferred antibody resistance in the context of BA.1 or BA.2 pseudotypes, but not in the

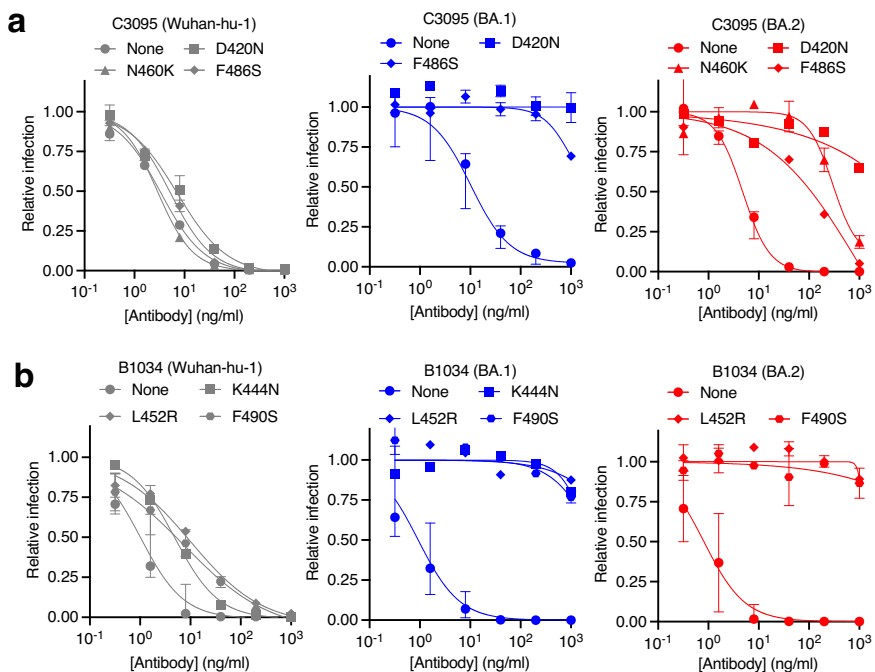

**Fig. 4 | Epistatic effects of RBD substitutions on neutralizing antibody sensitivity. a** Neutralization of the D420N, N460K, and F486S RBD point mutant pseudotypes in Wuhan-Hu-1, BA.1, and BA.2 backgrounds by antibody C3095. **b** Neutralization of the K444N, L452R, and F490S RBD point mutant pseudotypes indicated in Wuhan-Hu-1, BA.1, and BA.2 backgrounds by antibody B1034. The median value is plotted and the error bars indicate the range of 1–3 independent titrations as detailed in the accompanying source data files.

context of Wuhan-Hu-1 pseudotypes. Thus, substitutions introduced in the BA.1 and BA.2 contexts led to escape from a greater number of antibodies than the same substitutions in the Wuhan-Hu-1 context (Fig. 2, Supplementary Fig. 2). For many of these substitutions, we confirmed their effects by determining full neutralization curves using Wuhan-Hu-1, BA.1 and BA.2 pseudotypes (Fig. 4a, b, Supplementary Figs. 3–5). For example, substitutions D420N, N460K, and F486S caused negligible changes in sensitivity to the class 1/2 antibody C3095 in the Wuhan-Hu-1 context but conferred complete resistance or large potency deficits (10- to 100-fold increased $IC_{50}$) in the BA.1 or BA.2 context (Fig. 4a). Similarly, for the class 3 antibody B1034, the K444N, L452R, and F490S substitutions conferred modest potency deficits for Wuhan-Hu-1 pseudotypes, but near-complete loss of neutralization in the BA.1 and/or BA.2 contexts (Fig. 4b).

Similar epistatic effects were evident for 14 of the 15 antibody resistance substitutions that were tested in all three pseudotype backgrounds (Fig. 2, Supplementary Figs. 3–5, Supplementary Data 4). Substitutions at positions N354, F375, D420, L441, N460, and F490 conferred resistance to one or more antibodies when introduced into BA.1 or BA.2 but did not do so when introduced into Wuhan-Hu-1. For 11 antibodies, single-amino-acid substitutions (at positions T345, R346, K444, Y449, N450, L452, F486 or N487) conferred resistance in a context-independent manner. However, the very same substitutions conferred resistance to 10 other antibodies when introduced into BA.1 and/or BA.2 but not when introduced into the Wuhan-Hu-1. Overall, there was extensive epistatic interaction between antibody resistance substitutions and variation in the BA.1 or BA.2 spike proteins (Fig. 2, Supplementary Data 4).

Included in the set of broadly neutralizing antibodies was C099 (class 1), for which a structure of the Fab:RBD interface has been determined[14]. While we failed to generate single-amino-acid resistance mutations in the context of Wuhan-Hu-1, C099 resistance was readily generated by single-amino-acid substitutions in the BA.1 and BA.2 contexts (Fig. 2, Supplementary Fig. 3, Supplementary Data 4). Indeed, two substitutions (at D420 and N460) within or proximal to the target

epitope (Fig. 5a) were necessary to confer resistance to C099 in the Wuhan-Hu-1 context[14], but each of these substitutions, as well as F486S or N487D, could individually lead to full C099 resistance in BA.1 and/or BA.2 (Fig. 2, Fig. 5a, Supplementary Fig. 3). Similarly, single target epitope substitutions at positions R346, K444 and L441 conferred resistance to C032[14], a clonal ancestor of the affinity-matured class 3 broadly neutralizing antibody, C080, but none conferred resistance to C080 in the context of Wuhan-Hu-1 (Fig. 2, Fig. 5b, Supplementary Fig. 4). Again, however, these single substitutions could generate C080 resistance in the BA.1 or BA.2 background. Overall, we conclude that BA.1 and BA.2 variants more readily escape broadly neutralizing antibodies such as C099 and C080 because they already possess a subset of the multiple target epitope substitutions that are required for antibody resistance (Fig. 5a, b).

### Epistatic interactions for mutations selected in vitro and naturally in BA.2 daughter lineages

Recently, SARS-CoV-2 sublineages derived from BA.2 have displaced BA.1 and BA.2[25–27] (https://gisaid.org). These sublineages have thus far exhibited varying degrees of prevalence, but most encode additional RBD substitutions, including L452Q (BA.2.12), G446S (BA.2.75, BA.2.75.2, XBB), N460K (BA.2.75, BA.2.75.2, XBB, BQ.1, BQ.1.1), K444T (BQ.1, BQ.1.1), L452R, F486V (BA.4, BA.5, BQ.1, BQ.1.1), F486S, F490S (XBB), and R346T (BA.4.6, BA.2.75.2, XBB, BQ.1.1) (Fig. 6a)[25,26]. These substitutions are identical to, or at the same positions as, substitutions that we found to exhibit epistatic interactions with BA.2 substitutions to confer resistance to multiple broadly neutralizing antibodies (Fig. 2, Fig. 4, Supplementary Figs. 3–5). In particular, we found that RBD substitutions at positions that changed during the BA.2 to BA.5 transition (L452 and F486) conferred resistance to antibodies of multiple classes (class 2/3 and 3 for L452R and classes 1, 1/2, and 1/4 for F486S). Notably, for 7 of 14 antibodies, resistance conferred by L452 or F486 substitutions was dependent on the prior acquisition of substitutions in BA.2 (Fig. 6b). Similarly, the N460K substitution that arose during the BA.2 to BA.2.75

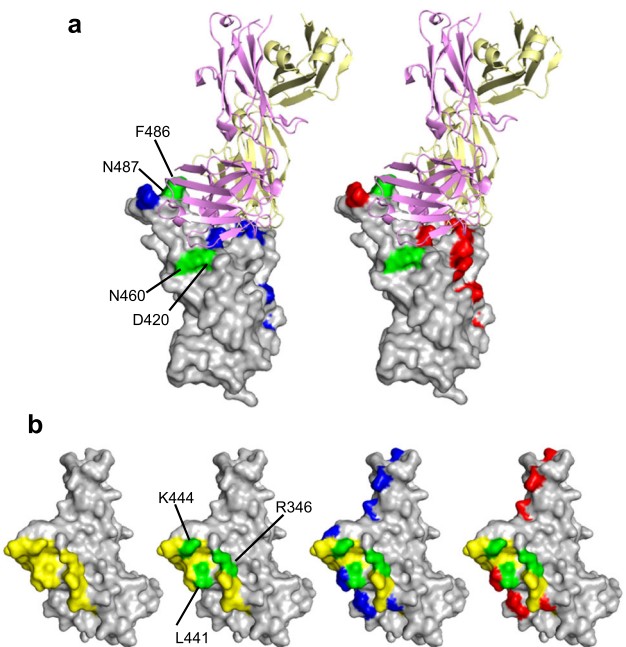

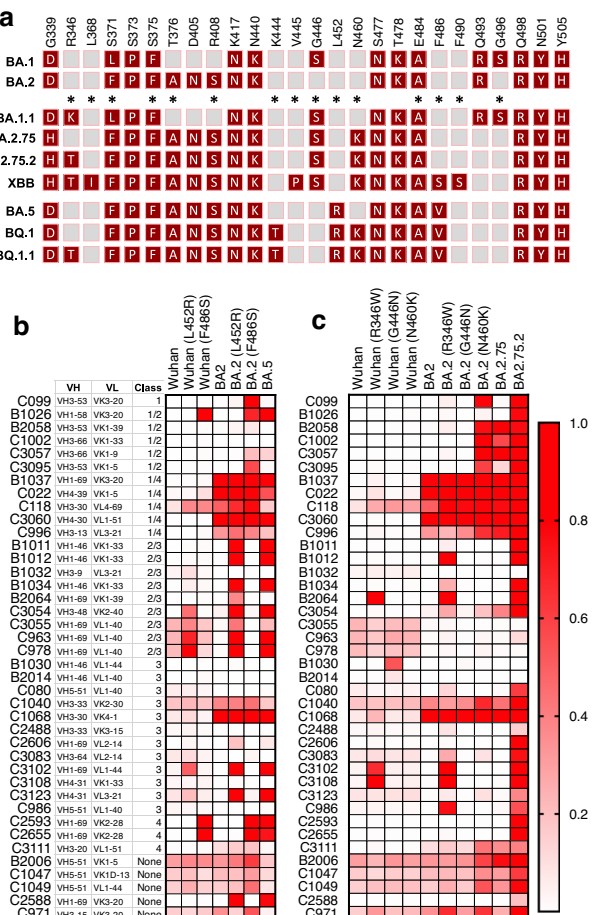

**Fig. 5 | Pre-existing and escape substitutions in broadly neutralizing class 1 and class 3 antibody epitopes. a** Substitutions that confer context-dependent escape from the class I antibody C099, depicted on the C099:RBD complex structure (PDB ID 7R8L[https://pdbj.org/mine/summary/7r8l]). Green indicates C099 escape substitutions. Blue and red indicate BA.1 and BA.2 substitutions, respectively. C099 heavy and light chains are magenta and yellow respectively. **b** Substitutions that confer context-dependent escape from C080 class 3 antibody, depicted on the Receptor-Binding Domain (RBD) complex structure (PDB ID 7C8J [https://www.rcsb.org/structure/7c8j]). Yellow indicates the C032 (clonal ancestor of C080) epitope. Green indicates escape substitutions. Blue and red indicate BA.1 and BA.2 substitutions, respectively.

**Fig. 6 | Context-dependent effect of BA.5, BA.2.75, and BA.2.75.2 substitutions on broadly neutralizing antibody sensitivity. a** Naturally occurring Receptor-Binding Domain (RBD) substitutions in a selection of BA.1, BA.2, and BA.5 derived SARS-CoV-2 variants. Positions marked with asterisks indicate those at which substitutions arose during broadly neutralizing antibody selection experiments. **b** Inhibition of Wuhan-Hu-1 and BA.2 RBD pseudotypes with L452R or F486S substitutions by broadly neutralizing antibodies, in comparison with BA.5. **c** Inhibition of Wuhan-Hu-1 and BA.2 RBD pseudotypes with R346W, G446N, or N460K substitutions by broadly neutralizing antibodies, in comparison with BA.2.75 and BA.2.75.2. **b, c** Relative infection is defined as the decimal fraction of infection measured (with 1 µg/ml antibody), relative to an uninhibited virus control (without antibody) and median values from two independent experiments are displayed.

transition conferred resistance to 6 class 1 or 1/4 antibodies with VH 3-53 and VH 3-66 heavy chains (Fig. 6c), that have previously been shown to generate antibodies that can neutralize BA.1 through BA.5 omicron variants[28]. Notably, the N460K substitution gave resistance to all 6 of the VH 3-53/VH 3-66 antibodies in the BA.2 background, but did not affect sensitivity to the same antibodies in the Wuhan-Hu-1 background (Fig. 6c). Cumulatively, substitutions at positions R346, G446, and N460 that arose during the BA.2 to BA.2.75 to BA.2.75.2 transitions, conferred resistance to 20 broadly neutralizing antibodies. However, substitutions at these positions individually conferred resistance to only four antibodies in the Wuhan-Hu-1 background (Fig. 6c). Overall, therefore, substitutions that occurred naturally in emergent SARS-CoV-2 variants exhibit substantial overlap with those selected by antibodies in vitro and exhibit clear context-dependent effects on antibody sensitivity.

### Resistance of synthetic SARS-CoV-2 variants to broadly neutralizing antibodies

Synthetic SARS-CoV-2 'polymutant' spike proteins, encoding combinations of changes arising during in vitro selection experiments, can have emergent escape characteristics similar to those of natural SARS-CoV-2 variants[3,7]. We generated synthetic RBD variants with combinations of mutations selected in vitro to assess the potential effect of selective pressure by broadly neutralizing antibodies. Specifically, we compiled RBD mutations that conferred resistance to non-overlapping subsets of antibodies (Fig. 2). For instance, the R346W substitution conferred resistance to several class 3 antibodies (e.g., C3102, C3108, C1068), the D420N substitution enabled escape from several class 1/4 antibodies (e.g., C099, C1002, C3095, C3111), while the K444N substitution enabled escape from various

antibodies, particularly in class 2/3 (e.g., B1011, B1012, B1034). In some cases, two or more alternative substitutions at spatially proximal positions (e.g., T345A and R346W or D420N and N460K) conferred resistance to similar sets of antibodies (Fig. 2). In these instances, only one substitution (R346W and D420N) was chosen. We tested 12 synthetic RBD variants, with 2 to 8 substitutions introduced in the context of a BA.5 spike (BA.5(+2) through BA.5(+8), Fig. 7a) for neutralization by the antibody panel. The synthetic variants all generated infectious pseudotypes, and all resisted neutralization by 32 or more of the 40 antibodies, across all classes (Fig. 7b). Notably, the BA.5(+2) and BA.5(+3a/b/c) variants, with only two (R346W, D420N) and three (R346W and D420N with K444N, V445E or P499S) substitutions, were resistant to 36 to 40 of the 40 broadly neutralizing antibodies. For one of the synthetic variants (BA.5(+3a)) we introduced the same three substitutions (R346W, D420N, and K44N) in the Wuhan-Hu-1 as well as the BA.5 contexts (Fig. 7b.) In the BA.5 context these three substitutions conferred resistance to 10 additional antibodies that they did not confer resistance to in Wuhan-Hu-1 context. Overall, the 40 broadly neutralizing antibodies inhibited

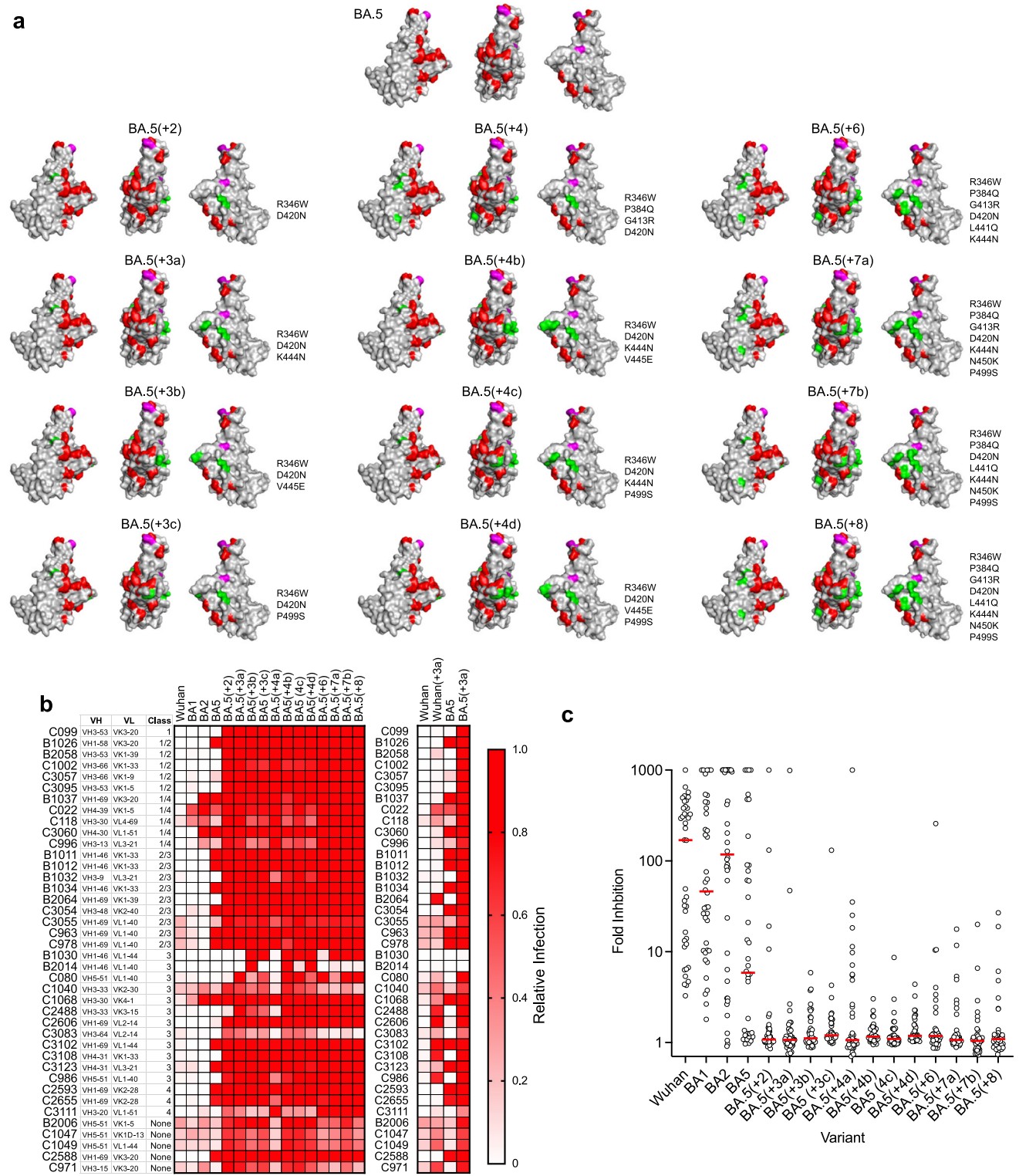

**Fig. 7 | Resistance of synthetic SARS-CoV-2 variants to broadly neutralizing antibodies. a** BA.5 and synthetic SARS-CoV-2 receptor-binding domain (RBD) variants. Red: substitutions in BA.2 RBD relative to Wuhan-Hu-1; magenta: substitutions BA.5 RBD relative to BA2. Highlighted in green are substitutions in the synthetic variants, chosen based on antibody selection experiments. **b** Inhibition of synthetic variants by broadly neutralizing antibodies. Relative infection was calculated as a decimal fraction of infection measured (with 1 µg/ml antibody) relative to an uninhibited virus control (without antibody). A median of two independent experiments is shown. **c** Fold inhibition of natural and synthetic variant infection by individual broadly neutralizing antibodies. Individual data points represent the median of two independent experiments for *n* = 40 individual antibodies. Red line = group median fold inhibition for all antibodies tested.

Wuhan-Hu-1, BA.1, and BA.2 pseudotype infection by a median of 170-, 40- and 118-fold at 1 µg/ml, respectively (Fig. 7c). For BA.5, the median inhibition was 6-fold, while for each of the synthetic variants, median inhibition was only 1.0- to 1.2-fold. We conclude that

compared to early omicron variants, none of the broadly neutralizing RBD antibodies examined herein remain effective against potential future SARS-CoV-2 variants derived from BA.5 with a limited number of additional substitutions.

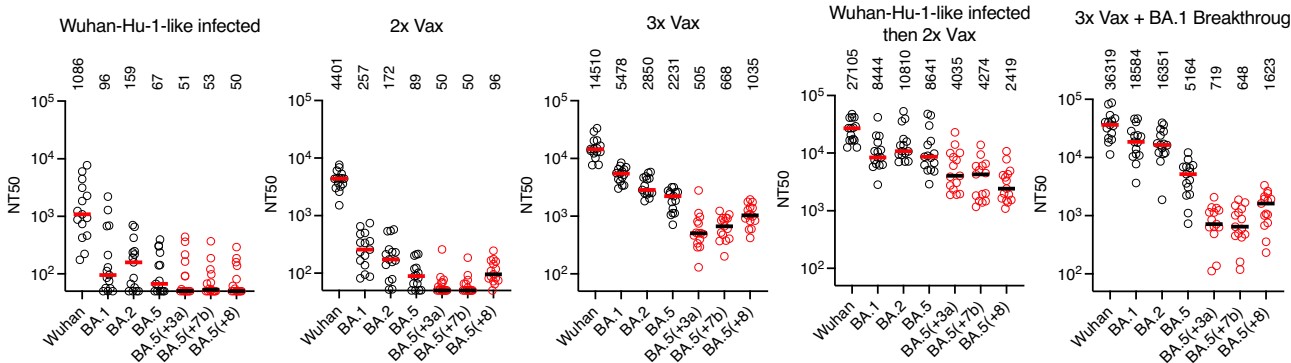

**Fig. 8 | Reduced potency of polyclonal plasma antibodies against synthetic SARS-CoV-2 variants.** The 50% neutralization titers ($NT_{50}$) for plasmas from various volunteer cohorts (see methods). Fifteen randomly selected plasma samples from each cohort were tested against natural variants; Wuhan-Hu-1, BA.1, BA.2, BA.5 (black), and synthetic variants; BA.5(+3) BA.5(+7b) BA.5(+8) (red). Individual points represent individual plasma samples ($n = 15$ for each cohort); median of two independent experiments is plotted. Group median $NT_{50}$ is indicated by red/black lines and numbers above each group.

## Plasma neutralization of synthetic SARS-CoV-2 variants

Populations of antibodies in the memory B-cell compartment may differ from those represented in plasma[29] and in contrast to the monoclonal antibody panel used herein, are not selected for particular properties. We, therefore, selected three synthetic variants (BA.5(+3a), BA.5(+7b), and BA.5(+8)) and determined their neutralization sensitivity to plasma antibodies from individuals with heterogeneous antigen exposures (Fig. 8, Supplementary Data 1, 5). Individuals who were infected with Wuhan-Hu-1-like viruses (but not vaccinated) exhibited low plasma $NT_{50}$ against all omicron variants, as did those whose only antigen exposure was two doses of an mRNA vaccine. These groups had median $NT_{50}$ titers of 96-257 for BA.1 and BA.2 variants and close to background $NT_{50}$ values for BA.5 and the synthetic BA.5-based variants (Fig. 8).

Plasma from three groups who had received multiple antigen exposures, (i) 3x vaccinated, (ii) infected then vaccinated, or (ii) 3x vaccinated and breakthrough infection—the same groups from which the broadly neutralizing antibodies were obtained (Supplementary Data 1)—had higher neutralizing titers against BA.1, BA.2 and BA.5 ($NT_{50}$ values of 2231–18584). The synthetic variants were less sensitive to neutralization by these plasmas than either of the BA.2 or BA.5 parental variants (Fig. 8). For example, plasmas from 3x vaccinated individuals, had median $NT_{50}$ values against BA.5(+3a) that were sixfold and fourfold reduced compared to BA.2 and BA.5 respectively. Notably, the group with BA.1 breakthrough infection following 3x vaccination had higher $NT_{50}$ values against BA.1 (3.4-fold) and BA.2 (5.7-fold) than those without breakthrough infection (Fig. 8). However, these apparent gains in neutralizing activity after BA.1 breakthrough infection were absent against the synthetic variants. Consequently, the BA.5 (+3a) variant was 23-fold and sevenfold less sensitive than BA.2 and BA.5 to plasmas from the BA.1 breakthrough infection group (Fig. 8). We conclude that a significant portion of the residual BA.5 plasma neutralizing activity in these groups can be evaded by the acquisition of a few additional substitutions that epistatically interact with pre-existing substitutions.

Notably, plasma from individuals who had been infected early in the pandemic and subsequently vaccinated (and were thus exposed to ancestral Wuhan-Hu-1-like antigens)[17] had broader neutralization properties than the 3x vaccinated and BA.1 breakthrough groups (Fig. 8). Indeed, for the infected then vaccinated group, the decrease in median $NT_{50}$ for BA.5 compared to BA.2 was not statistically significant, and the synthetic variants were only twofold to 3.6-fold less sensitive than BA.5 to these plasmas. These decrements reached significance only for the BA.5(+7b) and BA.5(+8) variants (p-values 0.02 and 0.01, respectively, Supplementary Data 5). While the $NT_{50}$ values for the infected then vaccinated versus the BA.1 breakthrough groups

were similar for Wuhan-Hu-1, BA.1 and BA.2 variants, plasma from the infected then vaccinated were significantly more potent against BA.5 and the synthetic variants ($p = 0.026$ to $p = 0.0016$, Fig. 8, Supplementary Data 5).

## Discussion

Prolonged or repeated exposure to antigen drives somatic mutation of antibody lineages resulting in greater diversity and affinity and, in the context of SARS-CoV-2 infection or vaccination, greater neutralization breadth and resilience to escape mutations[13–18]. A key consequence of antibody maturation is that SARS-CoV-2 escape from neutralization by individual RBD-specific antibodies often requires more than one substitution in the target epitope[14]. The mechanism underlying this effect is likely based on antibody affinity: as SARS-CoV-2 neutralizing antibodies evolve, they bind with higher affinity to their target epitope[14]. Thereafter, single-amino-acid substitutions may cause a reduction in binding affinity, which is insufficient to cause loss of neutralization activity for antibodies that bind with exceptionally high affinity[14]. A corollary, described herein, is that epistatic interactions between newly acquired and pre-existing viral substitutions enable escape from neutralizing antibodies that have a sufficiently high binding affinity that they tolerate single but not multiple target epitope substitutions. This feature is evident in antibody escape by BA.1 and BA.2 that encode many affinity-reducing mutations in class 1–4 RBD epitopes, which provide a background whereby a small number of additional RBD substitutions can result in additional affinity reductions that cause loss of neutralization for most of the broadly neutralizing antibodies tested.

Based on this premise, we posit that the rapid sequential displacement of early omicron variants by subsequent derivatives that are derived from BA.2, rather than from ancestral lineages, maybe in part because individual substitutions have greater impact on neutralization escape in omicron than in ancestral contexts, due to epistasis. Such a scenario is consistent with a transition to a SARS-CoV-2 phylogeny in which contemporary variants are derived from recent ancestors rather than a more distant ancestral root, which would typically characterize viral phylogeny in immunologically naïve populations[30]. Previously, studies with influenza virus have shown that one of two H3 hemaglutinin substitutions analyzed yielded context-dependent changes in neutralization sensitivity[31], suggesting that epistatic interactions between existing and emergent substitutions may generally drive neutralizing antibody escape in immunologically drifting viral populations.

Limitations of this study include the fact that the broadly neutralizing antibodies were isolated using a Wuhan-Hu-1 RBD bait to sort memory B cells, and selected for breadth using a BA.1-based neutralization assay. It is possible that antibodies selected using a different bait or selected for different neutralizing properties would have

yielded different results. Moreover, neutralizing antibodies that target other domains of the SARS-CoV-2 spike protein were not examined. However, the large decrement in polyclonal plasma neutralizing activity caused by the installation of a small number of RBD substitutions suggests that the activity of a large fraction of individual, affinity-matured, neutralizing antibody lineages are compromised by epistatic interactions between pre-existing and emergent RBD substitutions described herein.

Analysis of antibodies in plasma and in the Wuhan-Hu-1-selected memory B-cell compartment suggests that BA.1 breakthrough infection selectively boosts the subset of vaccine-elicited neutralizing antibodies residing in memory that neutralize BA.1[21,22,32,33]. While some of these memory antibodies also neutralized BA.5, most did not neutralize synthetic variants with only three additional substitutions. BA.1 infection after three vaccine doses thus provides no increase in plasma neutralization over three vaccine doses alone against such variants. In contrast, boosting of a diverse set of evolved antibodies, elicited by ancestral Wuhan-Hu-1 infection, using a Wuhan-Hu-1-based mRNA vaccine, increases and further matures, a broader set of antibodies[17]. Subsets of these antibodies may cross-react with distinct variants, as suggested by the ability of plasma from infected and then vaccinated individuals to neutralize the synthetic variants tested herein. Our findings thus suggest that differences in antigen exposure may generate significant heterogeneity in antibody-neutralizing breadth and potency. Differences in plasma-neutralizing potency against contemporary variants between groups with different antigen exposures are increasingly evident as the antigenic distance between currently circulating and ancestral SARS-CoV-2 variants grows. Overall, our studies indicate that epistatic interactions between pre-existing and new viral mutations, along with population heterogeneity in neutralizing antibody responses, will likely continue to impact the future direction and pace of SARS-CoV-2 variant emergence, with implications for vaccine effectiveness and selection of viral sequences for inclusion in updated vaccines.

## Methods

### Cell culture

All cell lines were cultured in Dulbecco's Modified Eagle Medium (DMEM) (ThermoFisher Scientific, 21013024) supplemented with 10% fetal calf serum and 10 μg/ml gentamicin at 37 °C and 5% $CO_2$. Cells were periodically checked for mycoplasma and retrovirus contamination by DAPI staining and reverse transcriptase assays, respectively. The cell lines used have been previously described[24]. Derivatives of 293 T and HT1080 cells expressing ACE2 were generated by transducing 293 T cells with CSIB(ACE2). Single-cell clones were derived by limiting dilution from the bulk populations and clones designated 293 T/ACE2cl.22, and HT1080/ACE2cl.14 were used in this study.

### SARS-CoV-2 variant spike expression plasmids

Generation of an omicron BA.1 spike expression plasmid lacking the C-terminal 19 amino acids and encoding the R683G furin cleavage site substitution (pCR3.1 BA.1Δ19) has been previously described[3]. The Omicron spike coding sequence was derived from sequence ID EPI_ISL_6640919. It was codon-optimized and synthesized as a C-terminally truncated Δ19 form in nine fragments (IDT). We also introduced a furin cleavage site mutation (R683G) that does not change the neutralization properties of the SASR-CoV-2 spike protein but enables higher titer pseudotyped viral stocks to be generated from transfected cells5. These synthetic DNA fragments, ranging in size from 444–599 bp and a NheI/XbaI-linearized pCR3.1 plasmid were Gibson assembled via 40 bps overlapping sequences. Individual plasmid clones were completely sequenced (Illumina MiSeq) and a single correct clone was used in these studies.

Plasmids encoding similar spike proteins from the BA1.1, BA.2, and BA.4/5 variants were derived from this plasmid by an overlap extension PCR strategy followed by Gibson assembly in the pCR3.1 plasmid that

was linearized by NheI/XbaI digestion. The variant-specific changes introduced were:

Omicron BA.1: A67V, Δ69-70, T95I, G142D, Δ143-145, Δ211, L212I, ins214EPE, G339D, S371L, S373P, S375F, K417N, N440K, G446S, S477N, T478K, E484A, Q493K, G496S, Q498R, N501Y, Y505H, T547K, D614G, H655Y, H679K, P681H, N764K, D796Y, N856K, Q954H, N969H, N969K, L981F.

Omicron BA.1.1: A67V, Δ69-70, T95I, G142D, Δ143-145, Δ211, L212I, ins214EPE, G339D, R346K, S371L, S373P, S375F, K417N, N440K, G446S, S477N, T478K, E484A, Q493K, G496S, Q498R, N501Y, Y505H, T547K, D614G, H655Y, H679K, P681H, N764K, D796Y, N856K, Q954H, N969H, N969K, L981F.

Omicron BA.2: T19I, L24S, del25-27, G142D, V213G, G339D, S371F, S373P, S375F, T376A, D405N, R408S, K417N, N440K, S477N, T478K, E484A, Q493R, Q498R, N501Y, Y505H, D614G, H655Y, N679K, P681H, N764K, D796Y, Q954H, N969K.

Omicron BA.5: T19I, L24S, del25-27, del69-70, G142D, V213G, G339D, S371F, S373P, S375F, T376A, D405N, R408S, K417N, N440K, L452R, S477N, T478K, E484A, F486V, Q498R, N501Y, Y505H, D614G, H655Y, N679K, P681H, N764K, D796Y, Q954H, N969K.

Oligonucleotide sequences used during molecular construction are given in Supplementary Data 6.

### Monoclonal antibodies and plasma samples

Monoclonal antibodies were cloned from the memory B cells of individuals with varying exposure to SARS-CoV-2 antigen as a result of infection and/or vaccination.

Plasma samples were from five groups

(i)   Individuals who were infected with Wuhan-Hu-1-like viruses -1 y prior to blood collection, but not vaccinated[17].

(ii)  Individuals who received two doses of mRNA vaccine (2nd dose -1 m prior to plasma collection)[34].

(iii) Individuals who received three doses of mRNA vaccine (3rd dose -1 m prior to plasma collection)[16].

(iv)  Individuals from the same Wuhan-Hu-1-like infection cohort who were infected -1 y previously and also received two mRNA vaccine doses prior to blood collection[17].

(v)   Individuals who received three mRNA vaccine doses and were then infected by omicron BA.1 -1 m prior to blood collection[21,22].

The study visits and blood draws were obtained with informed consent from all participants under a protocol that was reviewed and approved by the Institutional Review Board of the Rockefeller University (IRB no. DRO-1006, 'Peripheral Blood of Coronavirus Survivors to Identify Virus-Neutralizing Antibodies').

### rVSV/SARS-CoV-2/GFP construction and rescue

The generation of rVSV/SARS-CoV-2/GFP chimeric viruses encoding SARS-2 spike proteins has been previously described[24]. A plaque-purified variant designated rVSV/SARS-CoV-2/GFP$_{2E1}$ encoding D215G/R683G substitutions was used in this study. To introduce the spike proteins of VOCs BA.1.1 and BA.2, the respective spike sequences were amplified from the pCR3.1 constructs using PCR and primers specific for the VSV backbone. The rVSV backbone was digested with MluI/XhoI and the SARS-CoV-2 spike insert was introduced by Gibson assembly.

To rescue the recombinant viruses, HEK-293T/ACE2cl.22 cells were seeded $1 × 10^6$/well in poly-D-lysine-coated six-well plates 1 day prior to transfection. The following day, cells were rinsed with serum-free DMEM and infected with a recombinant vaccinia virus expressing T7 polymerase at an MOI of 5 for 45 min at 37 °C, gently rocking every 10–15 min. The inoculum was removed and 1.5 ml DMEM supplemented with 10% FCS was added. Next, cells were transfected with a plasmid mixture containing the rVSV/SARS-CoV-2 genome (500 ng) and the helper plasmids pBS-N (300 ng) (Kerafast, EH1013), pBS-P (500 ng) (Kerafast, EH1014), pBS-L (100 ng) (Kerafast, EH1014) and pBS-G

(800 ng) (Kerafast, EH1015) using the lipofectamine LTX (9 μl) (ThermoFisher, 15338100) and PLUS (5.5 μl) (ThermoFisher, 15338100) transfection reagents. The DNA and PLUS reagent were mixed in 100 μl OptiMEM (ThermoFisher, 31985070) to which LTX was added in 105 μl OptiMEM. The reaction was incubated for 20 min at RT and then added to the cells. The transfected cells were monitored by microscopy and the supernatant of cells expressing GFP was collected ~48 h post transfection, filtered using a 0.1 μm filter to remove residual vaccinia virus, and transferred to HEK-293T/ACE2cl.22 cells in six-well plates to amplify the rescued virus. After ~48 h the supernatant of the infected cells was collected and filtered using a 0.22 μm filter and transferred onto HEK-293T/ACE2cl.22 cells were seeded in T175 flasks. The viruses were passaged several times in T175 flasks to create diversity in the viral population and generate virus stocks of >10$^8$ IU/ml. The titer of the viral stocks was assessed by serial dilution, infection of HEK-293T/ACE2cl.22 cells, and subsequent flow cytometry (see Supplementary Fig. 6)

### Antibody sequencing, cloning, and expression

Antibodies were identified and sequenced[23]. In brief, RNA from single cells was reverse-transcribed (SuperScript III Reverse Transcriptase, Invitrogen, 18080–044) and the cDNA was stored at −20 °C or used for subsequent amplification of the variable IGH, IGL, and IGK genes by nested PCR and Sanger sequencing. Sequence analysis was performed using MacVector. Amplicons from the first PCR reaction were used as templates for sequence- and ligation-independent cloning into antibody expression vectors. Recombinant monoclonal antibodies were subsequently produced and purified[23]. Paired heavy and light chain expression constructs were transfected into 293-6E cells (NRC) using branched polyethylenimine (PEI) 25 kDA (Sigma). After 7 days of culture, cells were centrifuged at 4200 x g for 40 mins at 4 °C, and supernatants were filtered through 0.22 μM aPES (Thermo Nalgene Rapid-Flow). Antibodies were then purified from filtered supernatants using Protein G Sepharose 4 Fast Flow (GE Healthcare) according to standard protocols. Antibodies were buffer exchanged and concentrated into PBS using Amicon Ultra centrifugal filter (Millipore) with either a 30 or 50 kDA molecular weight cutoff.

### Biolayer interferometry

Biolayer interferometry assays were performed as previously described[23]. In brief, we used the Octet Red instrument (ForteBio) at 30 °C with shaking at 1000 r.p.m. Epitope binding assays were performed with protein A biosensor (ForteBio 18-5010), following the manufacturer's protocol "classical sandwich assay" as follows: (1) Sensor check: sensors immersed 30 sec in buffer alone (buffer ForteBio 18-1105), (2) Capture 1st Ab: sensors immersed 10 min with Ab1 at 10 μg/mL, (3) Baseline: sensors immersed 30 sec in buffer alone, (4) Blocking: sensors immersed 5 min with IgG isotype control at 10 μg/mL. (5) Baseline: sensors immersed 30 sec in buffer alone, (6) Antigen association: sensors immersed 5 min with RBD at 10 μg/mL. (7) Baseline: sensors immersed 30 sec in buffer alone. (8) Association Ab2: sensors immersed 5 min with Ab2 at 10 μg/mL. Curve fitting was performed using the Fortebio Octet Data analysis software (ForteBio). Affinity measurements of anti-SARS-CoV-2 IgGs binding were corrected by subtracting the signal obtained from traces performed with IgGs in the absence of WT RBD. The kinetic analysis using protein A biosensor (as above) was performed as follows: (1) baseline: 60 sec immersion in buffer. (2) loading: 200 sec immersion in a solution with IgGs 10 μg/ml. (3) baseline: 200 sec immersion in buffer. (4) Association: 300 sec immersion in solution with WT RBD at 20, 10, or 5 μg/ml (5) dissociation: 600 sec immersion in buffer. Curve fitting was performed using a fast 1:1 binding model and the Data analysis software (ForteBio). Mean KD values were determined by averaging all binding curves that matched the theoretical fit with an R2 value ≥0.8. To classify antibodies, competition with antibodies C105 (class 1) C144 (class 2) C135 (class 3), and CR3022 (class 4)[8,23] was determined.

### Selection of antibody-resistant rVSV/SARS-CoV-2 variants

To identify antibody escape mutations in rVSV/SARS-CoV-2/GFP$_{2E1}$, rVSV/SARS-CoV-2/GFP$_{BA1.1}$, and rVSV/SARS-CoV-2/GFP$_{BA2}$, viral populations containing $1 \times 10^6$ infectious units were incubated with monoclonal antibodies (mAbs) at a concentration of 1 μg/mL for 1 h at 37 °C. The virus/antibody mixture was then used to inoculate HEK-293T/ACE2cl.22 cells in 6-well plates. The next day, the medium was replaced with fresh medium containing the monoclonal antibody at 1 μg/ml. After a further 24 h, the virus-containing supernatant was harvested and passed through a 0.22 μm 96-well filter plate. The filtered supernatant (100 μL) was added to medium containing mAbs to achieve a final mAb concentration of 1 μg/ml in a total volume of 1 ml. The virus:antibody mixture was incubated for 1 h at 37 °C and used to inoculate $2 \times 10^5$ HEK-293T/ACE2cl.22 cells for a second passage (p2) in the presence of antibodies. Medium was again replaced with fresh antibody-containing medium after 24 h, and the putatively selected p2 virus population was harvested after 48 h. RNA was then extracted from 100 μL of filtered p2 supernatant and reverse-transcribed using the SuperScript VILO cDNA Synthesis Kit (Thermo Fisher Scientific). Sequences encoding the extracellular domain of the spike protein were initially amplified using KOD Xtreme Hot Start Polymerase (Sigma-Aldrich, 719753). In a subsequent PCR reaction, RBD-specific primers were used to amplify the receptor-binding domain (RBD), and the resulting products were purified and sequenced.

### Receptor-binding domain sequencing

PCR products encoding the RBD were subjected to tagmentation reactions, that included 0.25 μl of Nextera TDE1 Tagment enzyme (Illumina, 20060059), 1.25 μl of TD Tagment buffer (Illumina, 20060059), and 10 ng of RBD PCR product. The tagmentation reaction was performed at 55 °C for 5 min. Next, 3.5 μl of KAPA HiFi HotStart ReadyMix (Roche, 07958935001) and 1.25 μl of i5/i7 barcoded primers were added to the tagmented DNA to incorporate a unique indexing sequence. Following purification with AmpureXP beads (Beckman Coulter, A63882), the barcoded DNA from several reactions was pooled together and denatured with 0.2 N NaOH. The barcoded library was then diluted to 12 pM and sequenced using Illumina MiSeq Nano 300 V2 cycle kits (Illumina, MS-103-1001). Illumina sequencing reads were trimmed using the BBDuk function of Geneious Prime to remove adapter sequences and low-quality reads. The forward and reverse sequencing reads of each reaction were then aligned to the corresponding reference sequence and annotated for the presence of mutations. To detect RBD escape mutations, the minimum variant frequency (i.e.-the minimum fraction of reads that contained a substitution for it to be denoted as such) was set at 10%.

### Mutant spike expression plasmids and pseudotyped virus generation

Selected point mutations that arose in the rVSV/SARS-CoV-2 selection experiments were introduced into pCR3.1-based SARS-CoV-2 spike expression plasmids using a primer-based overlap extension approach as described above. Mutations that were selected in rVSV/SARS-CoV-2/GFP$_{2E1}$ populations were introduced into the corresponding pCR3.1-based Wuhan-Hu-1 spike expression plasmid. Conversely, mutations that were selected in rVSV/SARS-CoV-2/GFP$_{BA1.1}$, or rVSV/SARS-CoV-2/GFP$_{BA2}$, were introduced in the context of all three (Wuhan-Hu-1 BA.1, BA.2) spike expression plasmids.

The generation of (HIV/NanoLuc) SARS-CoV-2 pseudotyped particles was described previously[24]. In brief, HEK-293T seeded in 10 cm dishes were transfected with 2.5 μg of a pCR3.1-SARS-CoV-2 spike expression plasmid and 7.5 μg of the pHIV-1$_{NL4-3}$ ΔEnv-NanoLuc reporter virus plasmid using 44 μl PEI (1 mg/ml) in 500 ml serum-free DMEM medium. The medium was changed 24 h post transfection. The virus-containing supernatant was harvested at 48 h post transfection, passed through a 0.22 μm pore-size polyvinylidene fluoride syringe

filter, aliquoted, and stored at −80 °C. The titers were assessed on HT1080/ACE2cl.14 cells seeded $10^4$ cells/well in 100 μl medium in black flat-bottom 96-well plates. A fivefold serial dilution was performed and 100 μl was transferred to the target plate. The plates were incubated for 48 h and NanoLuc luciferase activity was determined as described below.

### Neutralization assays
To assess the neutralization of pseudotyped HIV-1 viruses encoding the RBD point mutations that we had identified during our selection experiments by mAbs, neutralization assays were performed. First, HT1080 cells were seeded at $10^4$ cells/well in 100 μl of medium. The next day, 60 μl of diluted virus (adjusted to give ~$10^7$ relative light units in the infection assay) were incubated with 60 μl of mAb solution for 1 h at 37 °C in a 96-well plate. Each mutant virus was initially screened for neutralization by the entire panel of monoclonal antibodies at a single antibody concentration of 1 μg/ml infection relative to no-antibody controls determined. For viruses deemed to be resistant to antibody neutralization, defined as having a relative infection value >0.4 relatives to the uninhibited virus control, $IC_{50}$ values were determined for parental virus and escape mutant. In this case, fivefold dilutions of the monoclonal antibody, starting at an initial concentration of 1 μg/ml, were incubated with the pseudotyped viruses. Thereafter, 100 μl of each antibody/virus mixture was added to the HT1080 cells and incubated at 37 °C for 48 h. NanoLuc luciferase activity was then determined.

### Reporter gene assays, and curve fitting
For NanoLuc luciferase assays, cells were washed twice with PBS and lysed with 30 μl of Cell Culture Lysis reagent (Promega, E1531). Nano-Luc luciferase activity was determined using the NanoGlo Luciferase Assay System (Promega, N1150). The substrate in NanoGlo buffer (30 μl) was added to the cell lysate. Luciferase activity was measured using a CLARIOstar Plus plate reader using 0.1 s integration time. The RLU readings were expressed as a decimal fraction of those derived from cells infected with pseudovirus in absence of antibodies.

To determine titers of rVSV/SARS-CoV-2/GFP, infected cells were trypsinized, fixed with 2% paraformaldehyde, resuspended in PBS/F-68, and the percentage of GFP-positive cells determined using an Attune NxT flow cytometer equipped with a 96-well CytKick Max autosampler.

Data were analyzed using Microsoft Excel for Mac (Version 16.65) and Prism V9.4.1 (GraphPad). $NT_{50}$ and $IC_{50}$ values were determined using a four-parameter nonlinear regression curve fit infectivity data measured as RLUs. The bottom values were set to zero, and the top values to one.

### Statistics and reproducibility
Statistical significance between groups was assessed by a two-tailed $t$ test using Welch's correction to account for unequal variances. All experiments were repeated successfully at least twice. No statistical method was used to predetermine the sample size. No data were excluded from the analyses; The experiments were not randomized. The Investigators were not blinded to allocation during experiments and outcome assessment, but the complexity of the experiments rendered the identification of single samples during experiments unlikely.

### Reporting summary
Further information on research design is available in the Nature Portfolio Reporting Summary linked to this article.

## Data availability
All source data for antibody escape mutation frequencies and for antibody neutralization information shown in the heatmaps are included in the supplementary data files 3 and 4.

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

## Acknowledgements

We thank Anna Gazumyan and Brianna Johnson for assistance with the expression of antibodies. This work was supported by National Institutes of Health (NIH) grant P01AI165075 to PDB, T.H., and M.C.N., R37AI64003 to P.D.B., R01AI78788 to T.H., P01-AI138398-S1, and 2U19AI111825 to M.C.N. F.M. was supported by the Bulgari Women & Science Fellowship in COVID-19 Research. P.D.B. and M.C.N. are Howard Hughes Medical Institute (HHMI) Investigators. This article is subject to HHMI's Open Access to Publications policy. HHMI lab heads have previously granted a nonexclusive CC BY 4.0 license to the public and a sublicensable license to HHMI in their research articles. Pursuant to those licenses, the author-accepted manuscript of this article can be made freely available under a CC BY 4.0 license immediately upon publication.

## Author contributions

L.W., V.A.B., and F.S. did VSV selection experiments. Z.W., A.C., R.R., F.M., F.S., T.H., cloned and chose antibodies for inclusion in the panel. L.W., V.A.B., C.G.-C., M.Canis., and D.J.P. did neutralization assays. C.G. and M.Caskey. recruited study participants. M.C.N., T.H., and P.D.B. supervised the study. L.W., V.A.B., and P.D.B. wrote the paper with input from other authors.

## Competing interests

The authors declare no competing interests.
