## [Peer Review File · Nature Communications]

Epistasis lowers the genetic barrier to SARS-CoV-2 neutralizing antibody escapeREVIEWER COMMENTS

Reviewer #1 (Remarks to the Author):

Dear Authors,

The manuscript entitled “Epistasis lowers the genetic barrier to SARS-CoV-2 neutralizing antibody escape” by Witte et al., is timely and will be of interest to the readers. The methods were designed meticulously and involved the replication competent recombinant vesicular stomatitis virus with SARS-CoV-2 spike protein (from Wuhan, Omicron-BA.1, and -BA.2) to study the escape mutant under the pressure of broadly neutralizing mAbs, followed by the generation of the pseudotyped lentiviral particles expressing the identified amino acid substitutions to the spike protein of Wuhan, BA.1, and BA.2 backbones for the neutralization assay. The findings were described in detail and highlighted that the epistatic interactions between newly emerging and the pre-existing amino acid mutations were required for SARS-CoV-2 to escape the neutralizing antibody.

There are some minor comments that I would like to address to this manuscript.

1. It is interesting to see that some amino acid substitutions, for example Y369F, render the virus more sensitive to the tested mAbs compared to the original version in both BA.1 and BA.2 backbones. Please elaborate more on this finding.
2. In Fig 2, Extended Fig 4-6, some data of the neutralization curves were omitted. To my understanding, authors might exclude the data of mutations that were not resistant to mAb. However, it would be better to still show all the test data side-by-side in all three backbones, i.e. Wuhan, BA.1 and BA.2.
3. Fig 2A and 3B, it would be useful for readers if authors incorporate the gene family of each mAb to the tables. The previous published data showed that many of the IGHV 3-53/ 3-66 mAbs managed to neutralize BA.1 through BA.5.
4. In Fig 4, it might be clearer to state “Wuhan-Hu1-like (or Early pandemic) infection” instead of just “infected” although this information was stated in the main text. It would also be better if authors can indicate the number of tested plasma samples from each cohort.
5. In the last paragraph before the discussion (corresponding to Fig 4), authors mentioned that the neutralization ability of samples obtained from individuals who had been infected by the early pandemic virus followed by subsequent vaccination had a broader neutralization properties compared to the other two cohorts (3 doses of vaccine and Omicron-BA.1 breakthrough). It would be great if authors also provide the statistical analyses comparing the data among groups, especially for the Omicron-BA.1, BA.2, BA.5 and BA.5 (+).
6. Please elaborate on the limitations of this study.

Best wishes,

Reviewer #2 (Remarks to the Author):

In this study, Witte et al. identified antibody escape mutants of ancestral Wuhan-Hu-1, BA.1, and BA.2 to 40 broadly neutralizing antibodies by passaging experiments. Interestingly, many antibodies could only be escaped by BA.1 and BA.2 but not Wuhan-Hu-1. Consistently, subsequent mutagenesis experiments showed that some escape mutations only worked for BA.1 and BA.2 but not Wuhan-Hu-1. These results demonstrate that the antigenicity of RBD can be influenced by epistasis. Using the information on those escape mutations, the authors generated a few mutants of BA.5 that could escape almost all 40 antibodies and showed stronger resistance to polyclonal plasma antibodies than BA.5.

Overall, the manuscript is timely, well written, and provides important insights into SARS-CoV-2

evolution. The conclusions are well-supported. I only have several minor suggestions to help improve the manuscript.

Minor comments:

1. The observation of differential antigenic effect of a mutation in different genetic background has been described for influenza hemagglutinin (PMID: 31017567), which should be cited in this study.
2. The mechanism of epistasis is unclear. Although investigating the mechanism is beyond the scope of this study, some speculations should be made in the discussion section.
3. For Figure 2a, it might be more informative to order the rows of the heatmap by antibody class as in Figure 3b.
4. Towards the end of the discussion section: “out” should be “our”.

Reviewer #3 (Remarks to the Author):

Witte et al. use SARS-CoV-2 pseudotypes to study potential pathways of SARS-CoV-2 spike protein resistance to monoclonal antibodies and polyclonal serum samples. They show that while certain point mutations in the spike protein receptor binding domain (RBD) have no effect when introduced into the ancestral spike protein sequence (Wuhan-Hu-1), the same single mutations result in significant antibody resistance when they are introduced into Omicron variants. They conclude that Omicron variants have a lower barrier to resistance to the monoclonal antibodies that remained effective against earlier Omicron variants. The study is timely, well-executed, and its findings are important to report.

1. Although affinities for the variants are not measured in this study, one would expect that most mutations would cause an effect on antibody affinity in the context of Wuhan-Hu-1, but that this change in affinity would not be large enough to impact neutralization. For example, if an antibody Fab binds the Wuhan-Hu-1 RBD with an affinity of 1 pM but binds BA.2 RBD with an affinity of 100 nM, then a single point mutation may have little effect on Wuhan-Hu-1 neutralization but could have a much larger effect on BA.2 neutralization. The term epistatic interactions is used to describe this phenomenon and the focus of the manuscript is neutralization, but some time should be spent explaining why this phenomenon may occur at the level of antibody affinities.
2. For the statement, “For example, class 4 or 1/4 broadly neutralizing antibodies, which bind epitopes that are well conserved in sarbecoviruses were escaped by mutations at P384, R408, G413, or G504 (Fig 2a, Extended Data Fig.2).” These mutations are seemingly not shown in Figure 2a?
3. As a related point, for Fig. 2a, most of the discussion on neutralizing monoclonal antibodies in the manuscript is focused on antibody classes rather than exposure status of the source donor. Could a legend/label be added to show what classes antibodies belong to?

REVIEWER COMMENTS (Responses in blue typeface)

Reviewer #1 (Remarks to the Author):

Dear Authors,

The manuscript entitled “Epistasis lowers the genetic barrier to SARS-CoV-2 neutralizing antibody escape” by Whitte et al., is timely and will be of interest to the readers. The methods were designed meticulously and involved the replication competent recombinant vesicular stomatitis virus with SARS-CoV-2 spike protein (from Wuhan, Omicron-BA.1, and -BA.2) to study the escape mutant under the pressure of broadly neutralizing mAbs, followed by the generation of the pseudotyped lentiviral particles expressing the identified amino acid substitutions to the spike protein of Wuhan, BA.1, and BA.2 backbones for the neutralization assay. The findings were described in detail and highlighted that the epistatic interactions between newly emerging and the pre-existing amino acid mutations were required for SARS-CoV-2 to escape the neutralizing antibody.

There are some minor comments that I would like to address to this manuscript.

1. It is interesting to see that some amino acid substitutions, for example Y369F, render the virus more sensitive to the tested mAbs compared to the original version in both BA.1 and BA.2 backbones. Please elaborate more on this finding.

Response: The revised manuscript includes a description and discussion of these data. Y369F falls within the class 4 antibody epitope, and it is the class 4 antibodies to which that substitution sometimes confers greater sensitivity. This could be due directly to changes in the antibody binding site, or due to increased exposure of the class 4 epitope.

2. In Fig 2, Extended Fig 4-6, some data of the neutralization curves were omitted. To my understanding, authors might exclude the data of mutations that were not resistant to mAb. However, it would be better to still show all the test data side-by-side in all three backbones, i.e. Wuhan, BA.1 and BA.2.

Response: We did not perform full neutralization curves for the majority of the antibody/pseudotype mutant virus combinations – this would have been an unfeasibly large amount of work. Rather, every antibody/pseudotype mutant was tested at a single Ab concentration (1µg/ml) and a representative sample of ‘escape’ mutants were then tested across a range of concentrations for a relevant antibody to verify that the larger set of single Ab dose/mutant combinations was yielding accurate findings – as can be seen from the data in Fig 2 and extended data 4-6 - this indeed proved to be the case.

3. Fig 2A and 3B, it would be useful for readers if authors incorporate the gene family of each mAb to the tables. The previous published data showed that many of the IGHV 3-53/ 3-66 mAbs managed to neutralize BA.1 through BA.5.

Response: In the revised version, we have re-ordered the antibodies in Fig 2a and 3b and included information on the gene family along with the neutralization heatmap data. We agree this makes the presentation clearer. The VH 3-53 and 3-66 constitute some, but not all, of the antibodies that neutralize BA.1 through BA.5. In particular, the VH 3-53 and 3-66 antibodies fall into the class 1 or class 1/2 groups, and their activity is abolished by epistatic interactions between preexisting BA.1/BA.2 mutations and the D420N or N460K mutations that arose in our selection experiments. Interestingly the newly emerging BA.2 and BA.5-derived subvariants BA.2.75, BA.2.75.2, XBB, BQ.1 and BQ.1.1 all contain the N460K substitution that confer resistance to this class of antibodies. We have expanded this portion of the results to incorporate this data (moving the previous Extended Fig. 8 to the main text to generate the new Fig. 3), and thank the reviewer for pointing this out.

4. In Fig 4, it might be clearer to state “Wuhan-Hu1-like (or Early pandemic) infection” instead of just “infected” although this information was stated in the main text. It would also be better if authors can indicate the number of tested plasma samples from each cohort.

Response the graphs have been modified to indicate “Wuhan-Hu-1-like infection” and the legend has been edited to indicate n=15 plasma donors for each cohort.

5. In the last paragraph before the discussion (corresponding to Fig 4), authors mentioned that the neutralization ability of samples obtained from individuals who had been infected by the early pandemic virus followed by subsequent vaccination had a broader neutralization properties compared to the other two cohorts (3 doses of vaccine and Omicron-BA.1 breakthrough). It would be great if authors also provide the statistical analyses comparing the data among groups, especially for the Omicron-BA.1, BA.2, BA.5 and BA.5 (+).

Response: We have included a statistical analysis of the data comparing the infected + vaccinated with 3x vaccinated and BA.1 breakthrough groups in the revised Supplemental Data 5 file, and highlighted some of these comparisons in the revised text

6. Please elaborate on the limitations of this study.

Response: We have included an additional paragraph in the discussion highlighting the limitations of the study.

Reviewer #2 (Remarks to the Author):

In this study, Witte et al. identified antibody escape mutants of ancestral Wuhan-Hu-1, BA.1, and BA.2 to 40 broadly neutralizing antibodies by passaging experiments. Interestingly, many antibodies could only be escaped by BA.1 and BA.2 but not Wuhan-Hu-1. Consistently, subsequent mutagenesis experiments showed that some escape mutations only worked for BA.1 and BA.2 but not Wuhan-Hu-1. These results demonstrate that the antigenicity of RBD can be influenced by epistasis. Using the information on those escape mutations, the authors generated a few mutants of BA.5 that could escape almost all 40 antibodies and showed stronger resistance to polyclonal plasma antibodies than BA.5.

Overall, the manuscript is timely, well written, and provide important insights into SARS-CoV-2 evolution. The conclusions are well-supported. I only have several minor suggestions to help improve the manuscript.

Minor comments:

1. The observation of differential antigenic effect of a mutation in different genetic background has been described for influenza hemagglutinin (PMID: 31017567), which should be cited in this study.

Response: Agreed, that study is cited in a revised manuscript in the context of a short discussion of the possibly general nature of the role of epistasis in viral immune escape.

2. The mechanism of epistasis is unclear. Although investigating the mechanism is beyond the scope of this study, some speculations should be made in the discussion section.

Response: We have added a section to the discussion to address this. Indeed, this critique is related to the point made by reviewer #3 – it is likely a ‘cumulative’ effect of the loss of binding affinity conferred by individual substitutions. Indeed, in our prior paper (Muecksh et al *Immunity* 2021) we found that affinity matured antibodies can indeed exhibit reduced affinity as a result of a single binding site substitution but retain neutralization potency. We’ve added a section to the discussion to make this point.

3. For Figure 2a, it might be more informative to order the rows of the heatmap by antibody class as in Figure 3b.

Response: Agreed, and reviewer 1 and 3 made a similar point. In the revised version, we have re-ordered the antibodies in Fig 2a and 3b and included information on the gene family and antibody class along with the neutralization heatmap data. We agree this makes the presentation clearer.

4. Towards the end of the discussion section: “out” should be “our”.

Response: Corrected in the revised version

Reviewer #3 (Remarks to the Author):

Witte et al. use SARS-CoV-2 pseudotypes to study potential pathways of SARS-CoV-2 spike protein resistance to monoclonal antibodies and polyclonal serum samples. They show that while certain point mutations in the spike protein receptor binding domain (RBD) have no effect when introduced into the ancestral spike protein sequence (Wuhan-Hu-1), the same single mutations result in significant antibody resistance when they are introduced into Omicron variants. They conclude that Omicron variants have a lower barrier to resistance to the monoclonal antibodies that remained effective against earlier Omicron variants. The study is timely, well-executed, and its findings are important to report.

1. Although affinities for the variants are not measured in this study, one would expect that most mutations would cause an effect on antibody affinity in the context of Wuhan-Hu-1, but that this change in affinity would not be large enough to impact neutralization. For example, if an antibody Fab binds the Wuhan-Hu-1 RBD with an affinity of 1 pM but binds BA.2 RBD with an affinity of 100 nM, then a single point mutation may have little effect on Wuhan-Hu-1 neutralization but could have a much larger effect on BA.2 neutralization. The term epistatic interactions is used to

describe this phenomenon and the focus of the manuscript is neutralization, but some time should be spent explaining why this phenomenon may occur at the level of antibody affinities.

Response: We agree that the likely mechanism underlying epistasis is a 'cumulative' loss of affinity that accrues through multiple mutations in the antibody binding site. Indeed, in our prior paper (Muecksh et al *Immunity* 2021) we found that affinity matured antibodies can indeed exhibit reduced affinity as a result of a single binding site substitution but retain neutralization potency. We've added a section to the discussion to make this point

2. For the statement, "For example, class 4 or 1/4 broadly neutralizing antibodies, which bind epitopes that are well conserved in sarbecoviruses were escaped by mutations at P384, R408, G413, or G504 (Fig 2a, Extended Data Fig.2)." These mutations are seemingly not shown in Figure 2a?

Response: this was an error – the data "callout" should have been (Supplemental Data 4, Extended Data Fig.1, Extended Data Fig.2).

3. As a related point, for Fig. 2a, most of the discussion on neutralizing monoclonal antibodies in the manuscript is focused on antibody classes rather than exposure status of the source donor. Could a legend/label be added to show what classes antibodies belong to?

Response: Agreed, and reviewer 1 and 2 made a similar point. In the revised version, we have re-ordered the antibodies in Fig 2a and 3b and included information on the gene family and antibody class along with the neutralization heatmap data. We agree this makes the presentation clearer.

Additional revisions

We have shortened the abstract to 150 words to conform to the *Nature Communications* Instructions to Authors. We also expanded and moved the old Extended Figure 8 to the main text (as the revised Fig. 3) to expand on the data showing that substitutions in naturally occurring BA.2-derived variants exhibit epistasis with pre-existing mutations. Finally, additional synthetic variants have been included in Fig. 4 which now demonstrates that all 40 broadly neutralizing antibodies can be escaped by a small number of substitutions in the BA.5 context.

REVIEWER COMMENTS

Reviewer #1 (Remarks to the Author):

The authors have responded and covered all the comments and suggestions. I have no further comment on this manuscript.

Best wishes,

Reviewer #2 (Remarks to the Author):

The authors have addressed all my previous concerns.